# A Simple Reversed Iontophoresis-Based Sensor to Enable In Vivo Multiplexed Measurement of Plant Biomarkers Using Screen-Printed Electrodes

**DOI:** 10.3390/s23020780

**Published:** 2023-01-10

**Authors:** Antonio Ruiz-Gonzalez, Harriet Kempson, Jim Haseloff

**Affiliations:** Department of Plant Sciences, University of Cambridge, Downing St., Cambridge CB2 3EA, UK

**Keywords:** ion-selective electrode, RuO_x_, pH sensor, non-invasive sensing, reversed iontophoresis

## Abstract

The direct quantification of plant biomarkers in sap is crucial to enhancing crop production. However, current approaches are inaccurate, involving the measurement of non-specific parameters such as colour intensity of leaves, or requiring highly invasive processes for the extraction of sap. In addition, these methods rely on bulky and expensive equipment, and they are time-consuming. The present work reports for the first time a low-cost sensing device that can be used for the simultaneous determination of sap K^+^ and pH in living plants by means of reverse iontophoresis. A screen-printed electrode was modified by deposition of a K^+^-selective membrane, achieving a super-Nernstian sensitivity of 70 mV Log[K^+^]^−1^ and a limit of detection within the micromolar level. In addition, the cathode material of the reverse iontophoresis device was modified by electrodeposition of RuO_x_ particles. This electrode could be used for the direct extraction of ions from plant leaves and the amperometric determination of pH within the physiological range (pH 3–8), triggered by the selective reaction of RuO_x_ with H+. A portable and low-cost (<£60) microcontroller-based device was additionally designed to enable its use in low-resource settings. The applicability of this system was demonstrated by measuring the changes in concentration of K^+^ and pH in tomato plants before and after watering with deionised water. These results represent a step forward in the design of affordable and non-invasive devices for the monitoring of key biomarkers in plants, with a plethora of applications in smart farming and precision agriculture among others.

## 1. Introduction

The design of non-invasive and precise methods for the monitoring of plant health has become a crucial field nowadays, showing promise for tackling the current challenges in food security. This technology could be used to reach the 70% increase in agricultural production needed by 2050 [1,2], by minimising the effects of environmental stress in plants. In particular, the presence of abiotic stresses such as extreme temperatures, light intensity, or drought can strongly reduce crop yield, leading to 50–70% losses [3]. In addition, biotic factors due to plant pathogens have been estimated to cause up to 16% production losses worldwide [4]. In the case of tomato plants, the second-most harvested crop worldwide, over 200 diseases and pests (i.e., fungi, bacteria, nematodes, viruses) have been reported [5]. However, in most cases, the diagnosis of plant pathologies is carried out by manual analysis from crop agronomists [6], being highly time consuming and lacking accuracy.

Hyperspectral imaging methods have become one of the most widespread technologies employed in the prevention of plant disease and stress [6]. This method consists of the measurement of emitted wavelengths of light by the plants. Typically, wavelengths within the visible, as well as the near-infrared range, are measured, allowing the determination of plant pigmentation [7]. However, these methods do not provide specific information about the dynamics of chemical biomarkers such as electrolytes or signalling molecules, which are essential for establishing an accurate diagnosis and prognosis of the plant. The use of pre-implanted nanomaterials inside leaves represents a step further in this field, by enabling the detection of molecules such as H_2_O_2_ and nitroaromatic compounds in real time [8]. However, these methods also lack selectivity, and require expensive equipment for their operation, limiting their use by the wider community of farmers.

The development of electrochemical methods for plant diagnosis has been proven to be a promising alternative to the use of hyperspectral imaging, reducing equipment costs and saving time due to their easy operation. This technology has been applied to the selective detection of a wide range of plant biomarkers, including phytohormones [9], glucose, and fructose [10], or even individual ions such as H^+^ [11]. In particular, the continuous measurement of cations such as K^+^ and Na^+^ could enable the early detection of diseases due to their crucial role in plant growth, as well as their involvement in the stress response and defence mechanisms of plants. K^+^ ions are essential for the optimal growth of plants, given their involvement in cell homeostasis and photosynthesis [12,13]. Deficiencies of this ion have been correlated with lower productivity yields [14], and higher susceptibility to diseases [15]. In addition, sap pH has been proven to be a powerful biomarker in the diagnosis of diseases such as the ones caused by Fusarium Oxysporum in Arabidopsis [16]. However, current approaches employed in the evaluation of plant K^+^ and pH rely on indirect measurements of electrolytes in soil, which cannot offer enough information about the physiological status of the plant. As such, new technologies for the non-invasive assessment of electrolytes in plants are needed, to enable the monitoring of plant health for smart agricultural applications.

Within the past few years, non-invasive electrochemical methods that can be used for the direct measurement of electrolytes in living organisms such as reversed iontophoresis have been developed and adapted for plant health monitoring. Reverse iontophoresis involves the extraction of analytes from living tissues using a low current, and the measurement of biomarkers by exploiting different electrochemical processes. Using this technology, González-Sánchez et al. [17] developed a sensor that could quantify the concentrations of plant hormones in leaves in a simple low-cost screen-printed electrode. This device could accurately determine salicylate concentrations within the millimolar level by applying cyclic voltammetry, using a voltage of between 0–1 V. However, the sensors were limited to the detection of one analyte, and the selectivity was poor. More recently, reversed iontophoresis has been used for the extraction of metabolites from leaves, which could be then identified using ion chromatography [18]. Although this system could determine the presence of multiple analytes, it required bulky and expensive equipment for the analysis of extracted compounds. Moreover, reversed iontophoresis has not been applied yet for the detection of electrolytes in plants.

In the present work, a low-cost and non-invasive sensing device has been designed for the simultaneous detection of K^+^ and pH. This sensor incorporated a screen-printed electrode where a plasticized PVC film had been deposited containing valinomycin as a K^+^-selective ionophore, and a pH-selective electrode based on RuO_x_. To enable its use in field applications, a low cost (<£60) and portable microcontroller-based device was additionally designed, which could perform reverse iontophoresis and ion detection processes simultaneously. The devices were calibrated initially in an aqueous environment. The real performance within a living plant was then tested by injecting different concentrations of electrolytes dissolved in DI water using a peristaltic pump. By applying a pulse voltametric/OCP hybrid approach, this system could be used for the simultaneous measurement of K^+^ and pH. The applicability of this approach was finally tested using a 6 week old tomato plant, allowing a determination of the changes in sap pH and K^+^ due to watering using DI water.

## 2. Materials and Methods

### 2.1. Materials

All reagents were purchased from Sigma Aldrich unless otherwise indicated: ruthenium (III) chloride, sodium alginate, high molecular weight chitosan, acrylamide, MBAA, acetic acid, high molecular weight poly (vinyl chloride), valinomycin, bis(2-ethylhexyl) sebacate, tetraphenyl borate, cyclohexanone, N,N′-methylenebisacrylamide, and N,N,N′,N′-tetramethylethylenediamine. Filter paper and all pH buffer solutions were purchased from Fisher Scientific. A Wio Terminal microcontroller was purchased from Seeed Studio. A peristaltic pump and resistors were purchased from Cool Components. A 10 µF capacitor was purchased from RS Components. Arduino pH-4502C pH meter was purchased from Morden Store. Screen-printed electrodes were purchased from Saida Technology. Compost was purchased from Mokuzai. Finally, tomato seeds (Gardener’s Delight variety) were purchased from Thompson and Morgan.

### 2.2. RuO_x_ Particle Synthesis and Characterisation

To allow the pH sensing, a RuO_x_ film was fabricated by electrodeposition onto the cathode electrode as reported elsewhere [19]. Briefly, a copper film was immersed in a solution containing 0.1 M RuCl_3_ in DI water (18 MOhm), and cyclic voltammetry was conducted for 20 min with an electrochemical station (Metrohm, Autolab BV, The Netherlands). A platinum electrode and an Ag/AgCl reference were employed, and a voltage between 0–0.8 V was applied at a rate of 0.1 V/s.

After the electrodeposition of RuO_x_ crystals, the morphology of the deposited film was studied by scanning electron microscopy (SEM, EVO LS15, ZEISS, Jena, Germany), using an accelerating voltage of 15 kV. In addition, the crystallinity of the RuO_x_ particles was determined by X-ray diffraction (XRD, Philips/PANalytical X’Pert PRO, Panalytical, UK). The XRD of pristine copper electrodes was first determined between 5–60°, and compared to the one obtained after the electrodeposition of RuO_x_ particles. The sensitivity of the material was finally measured electrochemically by using pulse voltammetry. This material was subjected to different pH solutions, created by increasing the pH of DI water using KOH. The acidity of the solution was then increased by using HCl. The pH of each solution was monitored using a pH meter (HI 2210 pH/C meter, Hanna Instruments, Leighton Buzzard, UK). After equilibrating the films on each pH solution for 2 min by recording the open circuit potential (OCP), a 0.8 V pulse was applied for 5 s, and the changes in current during the pulse voltammetry assay were recorded. The differences between the initial and final values of current could be correlated with the pH of the solution.

### 2.3. Hydrogel Synthesis for Incorporation onto Sensing Electrodes

The measurement of pH and K^+^ ions require an aqueous environment to allow the transport and adsorption of ions onto the electrodes. As such, to allow a portability of the device, enabling its use as a wearable sensor in leaves, two hydrogels were deposited onto the sensors and reverse iontophoresis electrodes. These hydrogels could preserve the moisture close to the electrode surfaces. A positively charged hydrogel was employed on the anode, where a positive voltage was applied to trigger the diffusion of anions from the leaf. This hydrogel consisted of cellulose/chitosan substrate. Initially, 2 wt% chitosan was dissolved in DI water containing 10 wt% acetic acid. A filter paper was then immersed for 1 h, and the final mixed material was dried overnight.

In the case of the iontophoresis cathode, where the cations diffuse during the reverse iontophoresis process, a negatively charged hydrogel was synthesized. This hydrogel was an alginate/polyacrylamide hybrid material adapted from the reported work by Yang et al. [20]. Initially, acrylamide monomers and sodium alginate were dissolved in DI water in 8:1 ratio. The final combined weight of both compounds reached 14 wt% in the DI water. Cross-linking agents N,N′-methylenebisacrylamide, (MBAA), ammonium persulphate (APS), and N,N,N′,N′-tetramethylethylenediamine (TEMED) accelerator were additionally dissolved to trigger the polymerisation reaction of acrylamide monomers and grafting to alginate using a molar ratio of 0.028 mol%, 0.031 mol% and 0.152 mol%, respectively. This polyacrylamide/alginate mixture was stirred at room temperature for 1 h, and a filter paper was immersed as the substrate. The final hybrid material was heated at 50 °C for 4 h and soaked in DI water. Both the chitosan and polyacrylamide/alginate hydrogels in filter paper were characterised by FTIR (L160000A Perkin Elmer, Waltham, MA, USA), evidencing the presence of the hydrogels in the paper substrate, and the formation of bonds between polyacrylamide and alginate.

### 2.4. Ion-Selective Electrode Fabrication and Characterisation

The final device incorporated a pH-sensing electrode, based on the previously described RuO_x_-modified copper, and an ion-selective electrode fabricated using a screen-printed system. The screen-printed electrodes incorporated a carbon-based counter electrode, an Ag/AgCl reference, and a working electrode where the ion-selective membrane had been deposited. The ISM was deposited using an aerosol method adapted from previous reported work [21]. The sensing films contained a standard film composition of 58 wt% DOS, 38 wt% PVC, 2 wt% valinomycin, and 2 wt% tetraphenyl borate. These compounds were dissolved in 10 mL of cyclohexanone and aerosolised using a pneumatic atomiser. 

After the deposition of ISM onto working electrodes, the morphology of the films in terms of roughness was determined by using a stylus profilometer (Dektakxt, Bruker, Coventry, UK). This method allowed the determination of surface homogeneity, which is essential for ensuring a good interface with the alginate/polyacrylamide hydrogels. The effects of incorporating this hydrogel film onto the ISM were additionally studied by calculating the changes in charge transfer resistance of the sensing films. These changes were determined by impedance spectroscopy, and simulating the results using an equivalent circuit. The impedance spectroscopy was conducted between 10^5^–10^−1^ Hz, with an amplitude of 0.01 V, and the used circuit was adapted from the work of Crespo et al. [22].

The sensitivity of the sensors in aqueous environments was finally determined. In this case, the sensors were immersed in aqueous solutions containing increasing concentrations of KCl (10^−8^, 10^−7^, 10^−6^, 10^−5^, 10^−4^, 10^−3^, 10^−2^, and 10^−1^ M, respectively). Initially, a voltametric pulse of 0.8 V was applied for 5 s. The open circuit potential (OCP) of the working electrode was then measured for 2 min, and the values after re-equilibration were used for the potentiometric calibration of the sensors. This potentiometric signal could be correlated with the logarithm of K^+^ concentration.

### 2.5. Assembly of Microcontroller-Based Device

To reduce the operation costs of the final sensing device, a microcontroller-based system was designed. This system incorporated a current measuring device and a circuit that allowed the application of a voltage and measurement of the resulting current. To allow the measurement of OCP from the screen-printed electrode, which could be correlated with the K^+^ concentration, an off-the-shelf device was adapted (pH-4502C, OCP sensor). In the case of the circuit used for the ion extraction and simultaneous pH sensing, a low-pass filter circuit with a 10 kOhm resistor in parallel with a 10 µF capacitor was employed. The current sensing was conducted by using a reference resistor (10 kOhm), which allowed a calculation of the produced current by extrapolation from the employed voltage (Appendix A). Both devices were connected to a Wio Terminal microcontroller, which could be programmed to automate the ion extraction and sensing processes. After the assembly of the devices, both pH and K^+^ sensors were calibrated using electrolyte solutions as described in previous sections. The devices were exposed to increasing concentrations of both KCl and HCl, and the results from the pulse voltammetry sensing approach were recorded.

### 2.6. Testing of Devices in Living Plants

Before the application of the wearable sensors for the determination of K^+^ and pH from living plants, the microcontroller-based devices were calibrated directly on a tomato plant, where the stem had been connected to a peristaltic pump. This method allowed a calibration of the sensing devices under a more representative scenario, and could be used to determine the differences in sensitivity due to the lower ion mobility of ions within living tissues compared to aqueous environments. Different KCl solutions were injected initially, at a flow rate of approximately 10 µL s^−1^. After the injection of each solution, the flow was established for at least 1 h before recording the potentiometric and amperometric results. To determine the pH sensitivity of the RuO_x_ films, different pH buffer solutions were used in this case, to minimise the interference effect from other cellular mechanisms that could trigger a change in sap acidity. In this case, leaves were gently wiped before the experiments were conducted, to avoid any interferences due to impurities present on the surface. The robustness of sensing devices was additionally demonstrated by calibrating the sensors using leaves with different textures (Hydrangea macrophylla, Capsicum annuum). In both cases, both the sensitivities towards pH and K^+^ concentration were determined.

The final test of the sensing devices involved the measurement of both K^+^ and pH in a 6 week old tomato plant. This plant had not been watered for 48 h prior to the experiment. Initially, the concentration of K^+^ and pH were determined in triplicate by our sensing device. The plant was then watered using 100 mL of DI water, which does not contain any ion. Both K^+^ and pH were then measured again 30 min after the plant had been watered.

## 3. Results and Discussion

### 3.1. Electrode Development and Characterisation

Within the present work, a flexible screen-printed electrode containing a Ag/AgCl reference and carbon-based counter and working electrodes was employed. The working electrode was modified by deposition of a plasticized PVC film containing valinomycin. This selective membrane was deposited by aerosol-assisted chemical deposition onto the carbon-based working electrode to ensure a good surface homogeneity and thickness control [21]. The final device presented a low roughness in the range of 15.21 nm (Appendix A) and could be used for the determination of potassium ions in solution. The surface roughness of ion-selective electrodes is key to ensure a proper performance of devices, since rough surfaces can increase the interfacial resistance of the sensing film due to a change in capacitive behaviour [21]. This increase in surface roughness has been shown to impact the stability and time response of the devices. Moreover, smooth surfaces are essential in ensuring the successful integration of a hydrogel upper layer, by avoiding the formation of air gaps between both films which allowed the operation of the device in wearable setups. As such, the use of aerosol deposition ensured the proper performance of K^+^ sensors in solution.

To ensure a proper diffusivity of charged ions within the electrode surface when used as a wearable device in plants, two different hydrogels were designed. These hydrogels were attached to the cathode and anode surfaces of the iontophoretic electrodes (Figure 1a). In the case of the anode, a cellulose/chitosan material was used to enhance the positive charge of the electrode. This material was developed by soaking a filter paper using a diluted solution of chitosan in 10 wt% acetic acid solution in DI water. On the contrary, an alginate/polyacrylamide hydrogel was employed in the case of cathode electrode, which was placed on the screen-printed device. This hydrogel was fabricated as reported by Hui et al. [20].

After the synthesis of both hydrogels, FTIR was conducted to ensure a proper composition of the films (Figure 1b). In the case of the chitosan/cellulose, the peaks at 3440 cm^−1^ and 3000 cm^−1^ corresponding to the -OH and -NH_2_ stretches were observed. The peak at 1600 cm^−1^ corresponding to the -CN stretches was additionally observed [23], indicating the presence of chitosan inside the cellulose-based substrate. The corresponding peaks from alginate and polyacrylamide within the hydrogel used in the cathode electrodes were additionally observed. In this case, the presence of a peak within the 3500 cm^−1^ due to the -OH groups in alginate and cellulose were present. In addition, the stretch from the carboxylate groups in alginate was observed at 1700 cm^−1^ [24]. These results indicate the successful synthesis of both hydrogels, which are essential for the ion sensing on plant leaves.

To allow a quantification of pH by the devices, the cathode electrode was modified using an electrodeposited RuO_x_ film. This material was selected given its specific reaction with H^+^ as described by the formula below [25]:$${\mathrm{RuO}}_{2}\cdot H_{2}O+H^{+}+e^{-}↔\mathrm{Ru}{\left(\mathrm{OH}\right)}_{3}\;$$

Ruthenium oxide thin films were fabricated onto copper electrodes by electrodeposition. This electrodeposition was conducted by cyclic voltammetry, using RuCl_3_ dissolved in DI water as the precursor material. A voltage of between 0–0.8 V at a rate of 100 mV/s was then applied (Figure 1c). This process led to the deposition of RuO_x_ particles onto the surface of the copper electrode, as evidenced by the obtained XRD pattern (Figure 1d).

Diffraction angles at 43° and 50° corresponding to the (1 1 1) and (2 0 0) of metallic Cu were observed in the case of pristine copper electrodes [26]. In addition, peak angles at 29° and 42° were present, being consistent with (1 1 0) and (2 0 0) planes from Cu_2_O [27], indicating the presence of an oxidised surface on the pristine copper electrodes. After modification with the deposition of RuCl_3_, peak angles at 28° and 56° were additionally obtained, being consistent with the presence of (1 1 0) and (2 1 1) planes of RuO_x_ [28]. Finally, the morphology of the electrodeposited RuO_x_ film was studied by SEM (Figure 1e), revealing the presence of RuO_x_ particles with a size in the range of 1 µm.

These results evidence the successful deposition of ruthenium oxide particles onto copper electrodes, which is essential for pH sensing. The combination of both ion-selective electrodes and ruthenium-oxide-based nanoparticles allowed the continuous and multiplexed sensing of pH and sodium ions in plants, which are crucial for plant physiology.

### 3.2. Device Electrochemical Testing

A pulse voltammetry/open circuit potential hybrid approach was adapted for the simultaneous measurement of pH and K^+^ ions. The use of a 0.8 V voltametric pulse allowed an extraction of ions from the plant leaves to the hydrogels (Figure 2a). In this case, the use of a positive voltage within the anode electrode propelled the extraction of negatively charged ions whilst positively charged ions such as H^+^ or K^+^ could diffuse towards the cathode. Both K^+^ and H^+^ ions were then measured attending to two different electrochemical mechanisms taking place at the anode: K^+^ ions could be selectively measured due to their specific chelation with the valinomycin ionophores contained in the ISM, triggering an increase in the OCP signal; on the contrary, H^+^ ions could react with RuO_2_ particles onto the copper electrodes, requiring electrons to form Ru(OH)_3_. Both processes could be recorded individually and used for the simultaneous measurement of the concentration. To avoid the accumulation of ions within the hydrogels, the polarity of the voltage was reversed by applying a positive voltage within the cathode electrode after every measurement. As such, the final devices could not determine both pH and K^+^ concentrations simultaneously, due to the different electrochemical processes required for the measurement. However, both parameters could be accurately measured within only a few seconds of operation of the sensors. Initially, during the 0.8 V pulse (5 s), the change in current was determined, and correlated with the pH, while the K^+^ showed a voltage of 0.8 V. Only after the voltage pulse had stopped could the OCP of the K^+^ electrode be measured and correlated with the concentration, while the pH sensors did not record any current.

To lower the operational costs and improve the portability of the system for field applications, an Arduino-based device was developed (Figure 2c). This device combined an off-the shelf open circuit potential meter, and an iontophoresis circuit for the application of a voltage pulse and measurement of the resulting current. The circuit employed for the reverse iontophoresis process is schematised in Appendix A.

The calibration of the sensing device was conducted by immersing the electrodes in different solutions containing increasing amounts of KCl. Initially, the open circuit potential of the modified screen-printed electrode was determined for 2 min. This signal was indicative of the electrochemical potential of the ISM and allowed the quantification of K^+^ ions in solution. A 0.8 V was then applied for 5 s, and the changes in the measured current were determined. These changes in the current were a consequence of the reaction between the RuO_x_ particles and H^+^, requiring an electron. As such, a decrease in the measured current was observed (Figure 2d). After the application of a voltametric pulse, the open circuit potential of the electrodes was recorded again for 2 min, until a stable baseline was reached. This sensing process could be successfully replicated using a portable microcontroller-based device, showing similar results for both pH and K^+^ sensing (Figure 2e).

When the measurement process was conducted in aqueous solutions containing KCl, the OCP signal after pulse voltammetry returned to the original baseline obtained before the pulse, indicating that the sensors were not polarised due to the application of a voltage. The effects of using a negatively charged hydrogel onto the PVC-based ISM were studied by impedance spectroscopy. This method allowed the measurement of charge transfer resistances by simulation using an equivalent circuit. In this case, the employed circuit was adapted from the work of Crespo et al. [22], and incorporated the contributions from solution resistance, charge transfer resistance, and film capacitance (Figure 3a). The measured charge transfer resistance of the pristine ISM was 3.92 ± 0.18 kΩ, increasing up to 13.72 ± 0.63 kΩ when the alginate/acrylamide membrane had been incorporated. On the contrary, the capacitance of the films decreased from 2.1 ± 0.1 nF up to 0.91 ± 0.04 nF. As such, the incorporation of an alginate/acrylamide film onto the PVC films lead to a small increase in the ion mobility. However, such an increase did not have an effect on the sensitivity or selectivity of the final devices.

The ion-selective electrodes were calibrated using multiple KCl solutions with increasing concentrations. The OCP of screen-printed electrodes could then be correlated with the concentration of K^+^ ions, showing a super-Nernstian sensitivity of 70.79 ± 10.11 mV Log[K^+^], and a detection limit within the micromolar level despite the presence of a hydrogel membrane on the electrode (Figure 3b). However, the amperometric sensitivity, calculated from the current obtained after the application of a voltametric pulse was relatively low, was in the range of 55 ± 19 µA Log[C]^−1^ (Figure 3c). This low change in current was attributed to the neutral pH found in KCl solutions. To study the selectivity of the devices towards K^+^ and pH, the sensors were additionally tested using increasing concentrations of KCl and KOH. The use of KCl allowed the calculation of the selectivity coefficient of the ISM film by the matched potential method. The measured value of selectivity was $k_{Na,\;K}^{Pot}$ = −2.2. This value was similar compared to electrodes employing the same ionophores and compositions such as the device reported by Han et al. [29] ($k_{Na,\;K}^{Pot}$ = −2.5). As such, the ISM contained within the screen-printed electrodes could be used for the selective determination of potassium ions in solution.

To study the effects of pH in the response of the sensors, the devices were subjected to several solutions containing different pH levels. To do so, the pH in DI was first increased up to 8 using NaOH. Increasing amounts of HCl were then added to reduce the pH down to 3. Both the OCP signal and current of the electrodes were then measured. A potentiometric sensitivity of 27.5 ± 6.7 mV pH^−1^ was obtained (Figure 3d), being low compared to the one achieved in the case of KCl (70.79 ± 10.11 mV Log[K^+^]). This result was consistent with the reported selectivity of the sensors towards K^+^. On the contrary, the amperometric sensitivity within the RuOx-modified copper electrode was significantly higher, reaching −1.3 ± 0.2 mA pH^−1^ (Figure 3e). This sensitivity was over 20 times higher than the one observed in the case of calibration using KCl and KCl, evidencing the sensitivity of the RuO_x_ films towards pH (Figure 3f).

The results obtained from the electrochemical testing of the sensing devices revealed a good potentiometric sensitivity and selectivity of the ISM towards K^+^. In addition, the RuO_x_ films demonstrated a high amperometric signal when the pH of the employed solution changed. However, despite the low-cost of the materials involved in the development of the sensing devices, the equipment needed for the measurement results is bulky and expensive, hindering its use within low-resource settings. Thus, to enhance the applicability of the sensing technology described in this manuscript, we developed a microcontroller-based device. The sensitivity of this low-cost system towards K^+^ and pH was tested using KCl and HCl as previously described. In this case, the potentiometric signal from the ISM onto the screen-printed electrode was recorded as the analog signal in the microcontroller device. A sensitivity of 96.3 ± 9.1 pt Log[K^+^]^−1^ was measured, while the linear range of the calibration remained within the micromolar range (Figure 4a). In addition, the sensitivity of the amperometric signal was low, in the range of −0.26 ± 0.46 µA Log[K^+^]^−1^ due to the negligible effect of KCl in the pH (Figure 4b). Conversely, when the pH was changed by the use of HCl, a similar behaviour of the microcontroller-based devices was observed compared to the results obtained by the electrochemical station. A low sensitivity of −15 pt Log[pH]^−1^ was recorded using the potentiometric signal (Figure 4c), while the amperometric sensitivity was in the range of −2.40 ± 0.22 µA pH^−1^ (Figure 4d).

Despite the lower amperometric response achieved in the case of the microcontroller-based device, the measured sensitivity was an order of magnitude larger than the one measured when the sensors were calibrated using KCl. As such, despite the lower sensitivity achieved using the Arduino-based device, the selectivity of the devices towards K^+^ and pH was preserved. These results indicate a good response of the sensing devices when they are calibrated in aqueous environments, which is favourable for their incorporation as wearable devices for plant monitoring. However, further testing was required to validate the accuracy of the sensing devices when applied onto plant leaf surfaces.

### 3.3. In Vivo Testing of Reverse Iontophoresis Device

The calibration of sensing devices in well-known electrolyte solutions showed a good response for both the pH and K^+^ sensing, even after the incorporation of a microcontroller-based system. However, due to the need of an extraction of ions from plant tissues, the observed sensitivity in liquid media could not accurately reflect the performance of the sensors when used on plant tissues. The diffusion of ions from living tissues during iontophoresis is highly dependent on the extraction time and applied voltage, as well as ion mobility in the medium [30]. As such, it is expected that the amount of extracted ions after each sensing cycle would be lower, due to the lower ion mobility of K^+^ ions inside plant leaves.

To ensure an accurate determination of the measured ion concentrations, the sensors were additionally calibrated on a plant, where a peristaltic pump had been incorporated within the stem (Figure 5a). This pump allowed the creation of a continuous flow of solutions containing known concentrations of electrolytes within the plant xylem. The sensing device, consisting of the screen-printed electrode and RuO_x_-modified copper contacts, was connected to a microcontroller for the measurement. The electrolyte solutions used in the previous section for the calibration of the screen-printed electrode were pumped through the plant stem for at least 1 h. Initially, the potentiometric performance of the sensing device was determined by injecting increasing multiple KCl solutions in the plant. The measured sensitivity was 76.8 ± 12.0 pt Log[K^+^]^−1^, being lower than the one previously recorded using DI water solutions (Figure 5b). In addition, the working range of the devices decreased, being in the range of 10^−3^–10^−1^ M. When the amperometric signal of the RuO_x_ film was calibrated, a sensitivity of −0.94 ± 0.10 µA pH (Figure 5c) was found. In this case, different pH buffers were employed for the calibration, ensuring a high control over the xylem pH.

This calibration using a real plant demonstrated the applicability of the reverse iontophoresis sensing device in the simultaneous determination of plant pH and K^+^ concentration. The obtained performance, with a limit of detection within the millimolar range, and good sensitivity towards both analytes, could be used for the monitoring of plant analytes in real time, allowing their incorporation for real-world applications in smart farming. Moreover, the sensors were robust, and could be deployed onto multiple leaves with different textures. This was tested by the calibration of the device using leaves from two different plant species (*Hydrangea macrophylla*, *Capsicum annuum*), achieving a similar sensitivity in all cases (Appendix A). However, the devices had to be calibrated to provide an accurate measurement. In the case of pH, sensitivities of −1.00 and −1.01 µA pH were achieved, being similar to the −0.94 ± 0.10 µA pH obtained in the case of tomato plants. The use of this low-cost sensing device in tomato plants will be explored within the next section.

### 3.4. Application of Sensing Device in the Monitoring of Plant Health Status

The reported sensor could determine the pH and K^+^ concentration in plant tissues with a detection limit within the millimolar level when used on a tomato leaf. The final sensing device showed a good flexibility, that allowed an easy incorporation onto the plant leaf (Figure 6a,b). In addition, to enable an easy and low-cost operation of these sensors, an intuitive interface was designed using a Wio Terminal device (Figure 6c), which enabled a monitoring of both pH and K^+^ in real-time, highlighting abnormal values in electrolyte concentrations. This microcontroller could also be used for the experimental design, allowing the user to change the supplied voltage.

To demonstrate the applicability of the sensing devices in the monitoring of K^+^ and pH in living plants, a full sensor containing the screen-printed electrodes and copper electrodes was placed onto a 6 week old tomato leaf. Both the potentiometric and amperometry signals were recorded using the calibration plots obtained within the previous section. The plant was then watered using DI water, and both signals were determined again (Figure 6d).

The calculated K^+^ concentration in plant leaves before watering was 5.48 ± 0.69 mM, being lower than the previously measured concentrations for tomato xylem sap [31,32]. These manuscripts report a concentration in the range of 20–30 mM. However, the methods employed in these studies are highly invasive, requiring cutting incisions on leaves, or involving hydraulic pumps for the extraction of sap. The formation of wounds due to pathogens or mechanical stimulation has been reported to trigger a rapid membrane potential change in vascular plants such as Arabidopsis [33]. In addition, an accumulation of cations such as Ca^2+^ and K^+^ has been reported in V. faba upon exposure to damage from Spodoptera littoralis [34]. As such, it is expected that the true concentration of K^+^ ions is overestimated in these studies, due to the wound signalling mechanism involved during the sap extraction process. Upon watering the plant with DI water, the measured potassium concentration decreased up to 0.19 mM, reflecting the lack of ions in DI water.

The pH of the plant also experienced a decrease, from 6.16 ± 0.64 up to 5.24 ± 0.31. Both pH lie within the physiological values reported in tomato plants [35]. In addition, a decrease in sap acidity has been reported in the case of xylem saps from plants under drought stress [36,37], being consistent with the observed results. This change in sap acidity is a consequence of the abscisic acid dynamic within the plant tissues when they are exposed to water shortage. Thus, the sensing device reported in this work could be used for the indirect analysis of water availability by the plant through pH sensing, as well as health status by the determination of K^+^.

Finally, the stability of the devices was tested over five days of operation. Although the K^+^ sensor provided a stable measurement throughout the five days of the experiment, the pH electrodes experienced a significant decrease in sensitivity after day 3 of measurements, reaching a value of 1.39. This decrease was attributed to a degradation of the RuO_x_ film due to electrochemical oxidation and dissolution as reported by Du et al. [38]. Such degradation increased the resistance of the films, reducing the measured current, and reduced the sensitivity of the sensors, leading to a lower measured pH value.

## 4. Conclusions

The present work reports for the first time a non-invasive sensing device that could be used for the monitoring of plant electrolytes using reversed iontophoresis. This sensor combined a K^+^-selective electrode modified with an alginate/acrylamide hydrogel, and a RuO_x_ particulate film that could be used for pH measurement. The final device employed a pulse voltammetry/OCP hybrid approach for the extraction and measurement of ions from plants. This method led to a super-Nernstian sensitivity of 70.79 ± 10.11 mV Log[K^+^], and good selectivity towards K^+^-ions when the screen-printed electrodes were used in aqueous solutions. In addition, the detection limit was within the micromolar range, showing promise for the accurate detection of this ion in biological solutions. The presence of RuO_x_ allowed the determination of pH between the range of 3–8, the typical range found under physiological conditions.

A low-cost microcontroller-based device was then designed. This system could successfully replicate the sensing approach adopted for the determination of K^+^ and pH, leading to similar detection limits in aqueous solutions. The practical application of this device in the detection of plant health status was demonstrated by incorporating the system onto a plant leaf. The sensing device could successfully determine pH and K^+^ concentrations in a living tomato plant sap, and record dynamic changes in both parameters. A decrease in both pH and K^+^ concentrations was detected after watering the plant with DI water, containing no ions. These measurements are likely to reflect the ion changes taking place on the leaf apoplast. As such, this work represents a step forward in the non-invasive monitoring of plant health, with a plethora of applications in smart farming and biological research among others. More importantly, the results obtained from the K^+^ measurements highlight the importance of developing portable and non-invasive methods for plant health assessment, to achieve an accurate understanding of plant–soil interactions and ion dynamics within the sap.

## Figures and Tables

**Figure 1 sensors-23-00780-f001:**
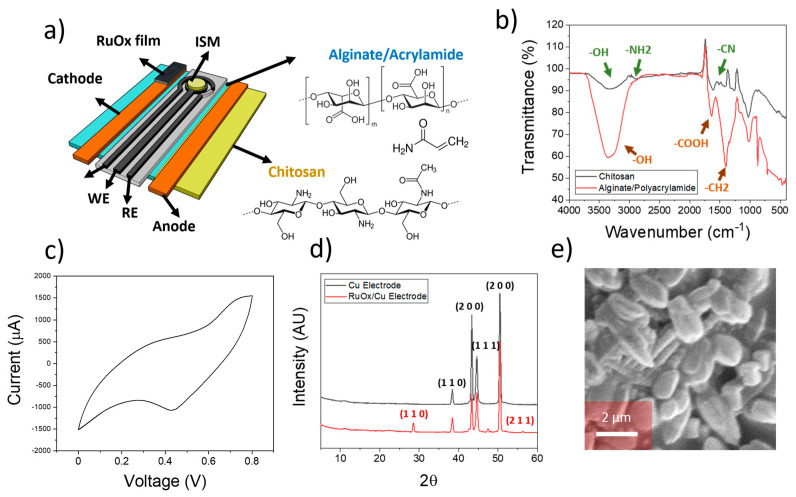
(**a**) Schematical representation of the device employed for the reversed iontophoresis experiments. The sensors included an ion-selective membrane within a three-electrode cell, and two different electrodes were incorporated for the ion extraction process. The different elements of the electrodes are indicated. (**b**) FTIR spectra of the hydrogels used within the anode (chitosan) and cathode (Alginate/Polyacrylamide) elements. (**c**) Cyclic voltammogram obtained during the electrodeposition of ruthenium oxide onto copper electrodes. (**d**) XRD plot of both copper electrode and RuO_x_-modified copper electrode used for the ion extraction process. The corresponding crystal planes from each element are indicated. (**e**) SEM imaging of electrodeposited RuO_x_ onto copper electrodes.

**Figure 2 sensors-23-00780-f002:**
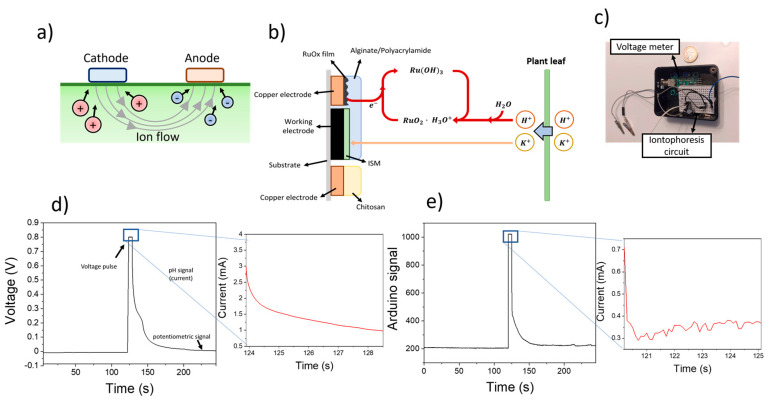
(**a**) Schematical representation of the reversed iontophoresis process, whereby the application of a voltage between the cathode and anode electrodes led to the extraction of cations and anions from the plant leaf respectively. (**b**) The sensing process is represented, involving the extraction of both K^+^ and H^+^ ions from the plant, which trigger an electrochemical reaction by the RuO_x_ film. (**c**) Picture of the microcontroller-based circuit used for the measurement of pH and K^+^ ions. (**d**) Plot showing the potentiometric changes of screen-printed electrodes before and after the use of 0.8 V for ion extraction. (**e**) Sensing plot obtained after using the Arduino-based device for the measurement.

**Figure 3 sensors-23-00780-f003:**
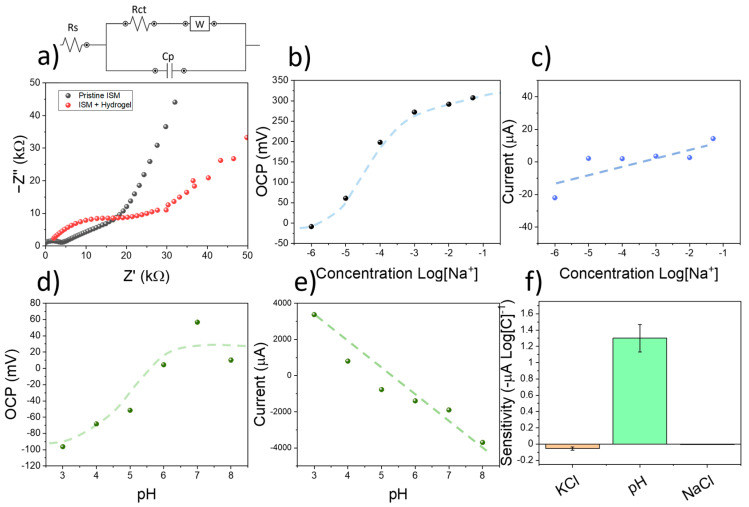
(**a**) Impedance spectroscopy plot of pristine ISM, and ISM where an alginate/acrylamide hydrogel film had been placed. The equivalent circuit employed for simulation is indicated. (**b**) OCP calibration plot of screen-printed electrodes using KCl. (**c**) Results from current obtained on the RuOx-modified copper electrodes during the measurement process. (**d**) OCP calibration plot obtained after calibrating the sensing devices using KOH. (**e**) The current obtained during the measurement process due to the applied voltage is indicated. (**f**) Comparison chart between the current sensitivities obtained at different concentration of electrolytes. The error bars represent the standard errors from the linear regression plots.

**Figure 4 sensors-23-00780-f004:**
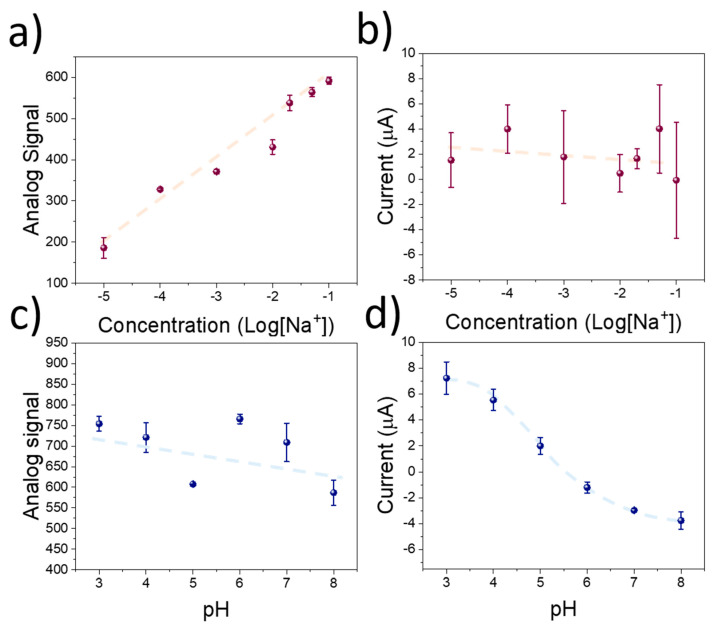
(**a**) Calibration plot of the microcontroller-based device using the screen-printed electrodes using different concentrations of KCl. (**b**) Amperometric signal recorded during the pulse voltammetry experiment using the RuO_x_-modified copper electrodes. (**c**) Potentiometric signal recorded during calibration of the sensing devices using KOH to increase the pH. (**d**) Corresponding amperometric signal recorded during the pulse voltametric experiment.

**Figure 5 sensors-23-00780-f005:**
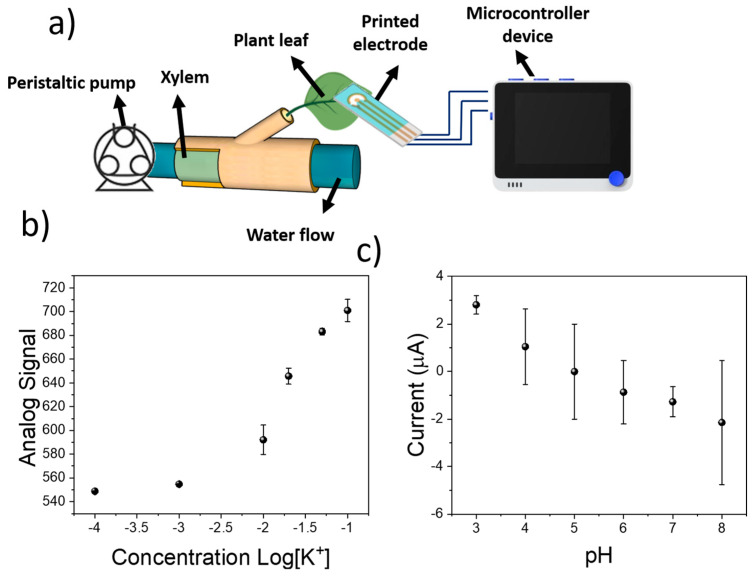
(**a**) Schematical representation of the system employed for the calibration of sensing devices in plants. The sensing electrodes were placed onto plant leaves and connected to a microcontroller for the sensing. (**b**) Calibration plot of screen-printed electrodes subjected to different KCl concentration in the plant xylem. (**c**) pH calibration of sensing devices by determination of current on RuO_x_ electrodes.

**Figure 6 sensors-23-00780-f006:**
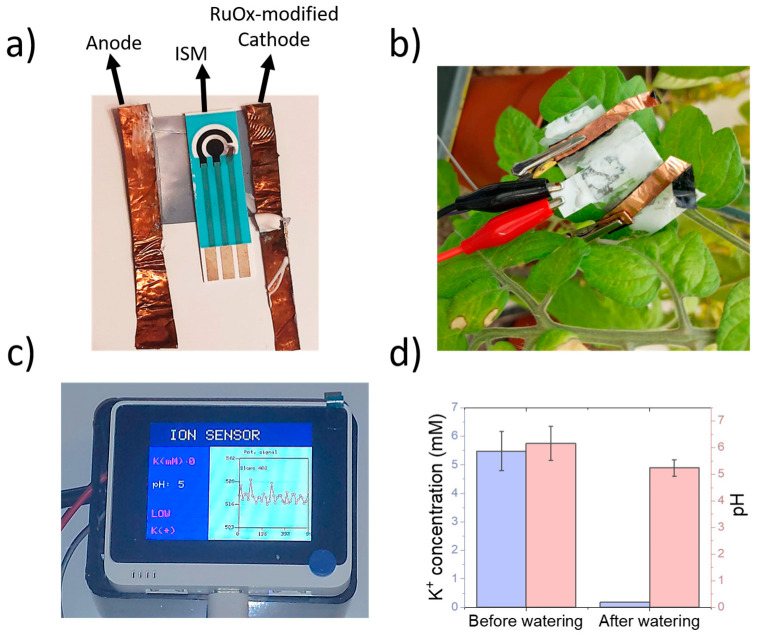
(**a**) Picture of sensing device, including a screen-printed three-electrode cell and copper electrodes for the iontophoretic extraction. Hydrogels were placed onto both electrodes and ISM. (**b**) The sensing device could be placed onto tomato leaves and connected to the microcontroller device using standard crocodile clips. (**c**) Designed user interface for experimentation and data visualisation using a microcontroller device. The potentiometric results are displayed in real time, and the average potassium and pH values are displayed. Non-healthy values of pH or K^+^ are displayed in red. (**d**) Results obtained during the measurement of both K^+^ and pH in tomato leaves before and after watering using DI water.

## Data Availability

The data presented in this study are available on request from the corresponding author.

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
