# Peer review of "A Simple Reversed Iontophoresis-Based Sensor to Enable In Vivo Multiplexed Measurement of Plant Biomarkers Using Screen-Printed Electrodes"

_sensors, 2023, doi:10.3390/s23020780_

Round 1
Reviewer 1 Report
This article entitled “A simple reversed iontophoresis-based sensor to enable in vivo multiplexed measurement of plant biomarkers using screen-3 printed electrodes” Manuscript ID: sensors-2092836, by Ruiz-Gonzalez et al. present a low-cost sensing device that can be used for the simultaneous determination of sap K+ and pH in living plants by means of reverse iontophoresis. The article demonstrates a step forward in the design of affordable and non-invasive devices for the monitoring of key biomarkers in plants which can be useful for smart farming technology. This article may be recommended for publication in the sensors after addressing few minor corrections.
Few Specific points:
1. Authors should clearly explain how the electrical connections are established here, simultaneous with one device; as both pH and ion sensor has different anode and cathodes. Can the sensor record both data simultaneously?
2. Line 228: “The final device presented a low roughness in the range of 15.21 nm (Figure S2.) and could be used for the determination of potassium ions in solution”
How profilometer data (Figure S2.) has been interpreted? It is not expressing the roughness 15.21 nm. Kindly explain how this roughness is related to the determination of potassium ions in solution?
3. Few Figure notations in the text is missing; Figure 1a; Figure 3e, Figure 4a and 4b, Figure 5b, …
4. Typos: Line 16 “from”; Line 11 “with time-consuming”; Line 114 “RuCl” repeated several times, mostly RuCl exits as RuCl3, kindly correct it all the text; Line 119 “RuO2”; Line 123 “5-60o; Line 145 “synthetized”; Line 177 “105-10-1 Hz”; Line 192 “ pH-4502C; Section 2.7 should be 2.6; Line 223 “This Within”; Line 261 “Figure 2C should be 1C”; Line 306 “Figure 2b should be 2c; Line 317 “Figure 2c should be Figure 2d”; line 329 “c” bold; Line 330 “d” bold; …..
5. All figures need improvement for better visibility, kindly increase the font size of X and Y axis in all figures
6. Figure 2d, 2e, 3a inset figures are not clearly visible.

Author Response
We would like to thank the reviewer for the positive comments on our work. We have addressed the minor revisions, and we hope that the manuscript is now up for publication:
- Authors should clearly explain how the electrical connections are established here, simultaneous with one device; as both pH and ion sensor has different anode and cathodes. Can the sensor record both data simultaneously?
We agree with the reviewer that a more in-depth explanation of the electrical connections is required. A detailed figure including all the electrical connections and sensors used has been included (Figure S2.), and it has been noted within the main text (P.7.)
Given the dual electrochemical process required for the determination of pH and potassium, involving a short (5 s) voltage pulse, and OCP measurements, both data cannot be recorded simultaneously. During the voltammetric pulse, a voltage of 0.8 V was imposed on the electrodes, and the current change was recorded to allow the quantification of pH. However, during this time, the voltage recorded by the K+ sensor was also 0.8 V. After the pulse, once the application of the voltage ceased, the OCP could be determined, and related to K+ concentrations. However, although both pH and K+ could not be determined simultaneously, their measurement only required a few seconds of operation. This has now been clarified within the main text (P.7).
- Line 228: “The final device presented a low roughness in the range of 15.21 nm (Figure S2.) and could be used for the determination of potassium ions in solution”. How profilometer data (Figure S2.) has been interpreted? It is not expressing the roughness 15.21 nm. Kindly explain how this roughness is related to the determination of potassium ions in solution?
We would like to apologize for the confusion of this statement. The roughness of ion-selective electrodes has been proven to impact on the performance of devices, since it can increase the interfacial resistance [1], which could reduce the sensitivity or time-response of the sensors. Moreover, in this work in particular, a high roughness could also be translated in a poor interface between the Alginate/Polyacrylamide hydrogel, used to keep moisture and attract cations within the vicinity of the ion sensor, since it could avoid the formation of air gaps. As such, keeping a low roughness was key to ensure a good long-term performance, and could be achieved using the aerosol method we developed as part of our previous work. This has been explained within the main text, and references has been included (P.5.).
[1] Ruiz-Gonzalez, A. and K.-L. Choy, Integration of an Aerosol-Assisted Deposition Technique for the Deposition of Functional Biomaterials Applied to the Fabrication of Miniaturised Ion Sensors. 2021. 11(4): p. 938.
- Few Figure notations in the text is missing; Figure 1a; Figure 3e, Figure 4a and 4b, Figure 5b, …
Figure notations have been indicated in the text.
- Typos: Line 16 “from”; Line 11 “with time-consuming”; Line 114 “RuCl” repeated several times, mostly RuCl exits as RuCl3, kindly correct it all the text; Line 119 “RuO2”; Line 123 “5-60o; Line 145 “synthetized”; Line 177 “105-10-1 Hz”; Line 192 “ pH-4502C; Section 2.7 should be 2.6; Line 223 “This Within”; Line 261 “Figure 2C should be 1C”; Line 306 “Figure 2b should be 2c; Line 317 “Figure 2c should be Figure 2d”; line 329 “c” bold; Line 330 “d” bold; …..
We would like to thank the reviewer for highlighting these typos. They have been corrected within the main text as indicated.
- All figures need improvement for better visibility, kindly increase the font size of X and Y axis in all figures
As suggested by the reviewer, all figures have been improved by increasing the font size of axis and titles.
- Figure 2d, 2e, 3a inset figures are not clearly visible.
Inset figures have been updated to improve the visibility. They have been moved out of the plots, so they could be expanded.
Reviewer 2 Report
This work reported a low-cost sensing device that can be used for the simultaneous determination of sap K+ and pH in living plants by means of reverse iontophoresis on screen printed electrodes. The work is interesting. However, there are still some issues in the manuscript that need to be improved. I suggest a minor revision for publication.
Comments:
1. Can the sensor achieve continuous monitoring over a long period of time? How long can the sensor work stably? It is recommended that relevant data be given.
2. Do different types of foliage affect the operation of the sensor? Some plants can have fuzz on the surface of their leaves. It is recommended that this feature be taken into account in practical application of sensing device.
3. The conclusions are too long and should be streamlined.
Author Response
We would like to thank the reviewer for the positive comments on our work, and indicating that our work is interesting. We have addressed the comments, and we hope that the manuscript is now up for publication:
Comments:
- Can the sensor achieve continuous monitoring over a long period of time? How long can the sensor work stably? It is recommended that relevant data be given.
Our devices could achieve a continuous and stable monitoring for at least 3 days. However, the sensors had to be re-humidified daily to avoid the drying of electrodes, which could change the measured signals. Although the measurements obtained by the K+ sensors were consistent, and showed a low variability during at least 5 days, the pH sensors decreased significantly after the third day of continuous operation, reaching values under 2. This could potentially be a consequence of the degradation of the RuOx film over time through electrochemical oxidation and dissolution as reported by Du et al. [1]. This has now been explained within the main text (P.14.), and a figure with the stability results across 5 days of operation has been included (Figure S.5.).
[1] Du, K., et al., Interface engineering breaks both stability and activity limits of RuO2 for sustainable water oxidation. Nature Communications, 2022. 13(1): p. 5448.
- Do different types of foliage affect the operation of the sensor? Some plants can have fuzz on the surface of their leaves. It is recommended that this feature be taken into account in practical application of sensing device.
We agree that different type of foliage could potentially interfere with the quantification. Throughout all the experiments, leaves were gently wiped, to ensure that potential debris that could interfere with the measurements. However, the presence of fuzz on the leaves could still represent a potential interference. To demonstrate the robustness of this approach, we have now conducted a calibration on 2 different plants, with different leaf textures (Hydrangea macrophylla, Capsicum annuum), and a similar sensitivity was obtained. This information has now been included in the main manuscript, within the materials and methods section (P.5.), and Results and Discussion (P.12.). Relevant calibration plots have also been added within the supplementary information (Figure S.4.).
- The conclusions are too long and should be streamlined.
As suggested, the conclusions section have been summarised, to highlight the key results.